# Three-Component Condensation of β-Ketonitriles, 4-Fluorobenzaldehyde, and Secondary Cyclic Amines

**Dmitry V. Osipov** [ID]**, Kirill S. Korzhenko and Vitaly A. Osyanin ***

Department of Organic Chemistry, Chemical Technological Faculty, Samara State Technical University,
244 Molodogvardeyskaya St., Samara 443100, Russia
* Correspondence: vosyanin@mail.ru

**Abstract:** A new three-component condensation of β-ketonitriles, 4-fluorobenzaldehyde, and secondary cyclic amines was developed. A possible reaction mechanism has been proposed including Knoevenagel condensation followed by aromatic nucleophilic substitution. It was found that in the case of 3-oxopropanenitrile bearing the 6-amino-1,3-dimethyluracil moiety, the reaction is not accompanied by fluorine substitution in the Knoevenagel adduct, and the Michael addition of a secondary amine occurs followed by oxidation.

**Keywords:** α-arylidene-β-ketonitriles; α-cyanoketones; three-component reaction; Knoevenagel condensation; aromatic nucleophilic substitution

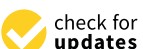



## 1. Introduction

The unflagging interest in the development of new multicomponent reactions is due to a number of their significant advantages compared to two-component reactions: a reduction in the number of synthetic stages, the simplicity and availability of reagents, simplification of the process of isolating final compounds, reduced solvent consumption, and, as a result, their environmental friendliness and higher efficiency. Multicomponent synthesis is often used in the complete synthesis of complex natural compounds and various carbo- and heterocyclic compounds. It requires a minimum set of initial substances and allows one to obtain entire libraries of compounds that have a structure similar to the biologically active components of drugs [1,2].

α-Arylidene-β-ketonitriles are important α,β-unsaturated compounds that are obtained via the Knoevenagel condensation reaction between β-ketonitriles (α-cyanoketones) and aryl aldehydes. So far, a broad range of works on the biological and pharmaceutical activities of β-carbonyl substituted nitriles has been published. For example, they have been recognized as anti-hyperglycemic (compound **I**) [3] and anti-tuberculosis (compound **II**) [4] agents; entacapone (compound **III**) [5] is a medication commonly used in combination with other medications for the treatment of Parkinson's disease. Furthermore, among α-arylidene-β-ketonitriles, a large number of potential cytotoxic agents (for instance, compounds **IV–VI**) have been identified (Figure 1) [6–8].

We were particularly interested in the synthesis of α-arylidene-β-ketonitriles due to their ability to be further functionalized. A high degree of polarization of the double carbon–carbon bond makes compounds of this type sensitive to both 1,3-dienes (Diels–Alder reaction) and nucleophiles (Michael reaction) [9–11]. α-Arylidene-β-ketonitriles are important building blocks that are used in the synthesis of various heterocycles [12] such as condensed 4H-pyrans [13,14], 2H-pyrans [15], and 3,4-dihydro-2H-pyrans [16]; 2,3-dihydrofurans [17,18] and furans [19]; 5,6-dihydro-4H-oxocines [20], dihydropyrimidines [21]; and pyridine derivatives [22] for the synthesis of quinolones, chromenes [23], functionalized 2,3-dihidroixazoles [24], 1H-pyrazolo[3,4-b]pyridines [25], and others, which are frequently found in pharmaceuticals and biologically active compounds. Therefore, the

development of new and more efficient methodologies for the synthesis of a wide variety of α-arylidene-β-ketonitriles has attracted a great deal of interest from synthetic organic chemists in past decades.

**Figure 1.** Representative examples of pharmacologically active β-carbonyl substituted α-arylidenenitriles.

## 2. Materials and Methods

### 2.1. Materials and Instrumentation

FTIR spectra were taken on a Shimadzu IR Affinity-1 spectrophotometer with a single-reflection ATR accessory and are reported in cm$^{-1}$. $^1$H, $^{13}$C, $^{19}$F NMR (400, 100, and 376 MHz, respectively) as well as DEPT-135 spectra were registered on a JEOL JNM-ECX400 spectrometer in DMSO-$d_6$ or CDCl$_3$, with the residual solvent signals (DMSO-$d_6$: 2.50 ppm for $^1$H nuclei, 39.5 ppm for $^{13}$C nuclei; CDCl$_3$: 7.26 ppm for $^1$H nuclei, 77.2 ppm for $^{13}$C nuclei) or CFCl$_3$ (0.0 ppm for $^{19}$F nuclei) serving as the internal standard. Chemical shifts and coupling constants were recorded in units of parts per million and hertz, respectively. High-resolution mass spectra (HRMS) were recorded on an Agilent 6230 TOF using an electrospray (ESI) ionization source. Melting points were determined by the capillary method on an SRS OptiMelt MPA100 apparatus. Monitoring of the reaction progress and assessment of the purity of synthesized compounds were conducted by TLC on Merck silica gel 60 F$_{254}$ plates, visualization under UV light and in I$_2$ vapor. All of the reactions were carried out in open air. Chemicals were purchased from the suppliers and used without further purification. Commercially unavailable β-ketonitriles **1** were prepared according to the procedure reported [26].

### 2.2. General Procedure for Preparation of α-Arylidenenitriles **4a–c,e–o** and Products **7a,b**, **9**

A mixture of the nitrile **1**, **5**, or **8** (1 mmol), 4-fluorobenzaldehyde **2** (124 mg, 1 mmol), and cyclic secondary amine **3** (2 mmol) in 3 mL of acetonitrile was heated at boiling and stirring for 6 h. The reaction mixture was cooled to –30 °C, the precipitate formed was filtered off and washed with ice-cold methanol. In the cases where there was no precipitation, the reaction mixture was concentrated under reduced pressure and the residue was purified by recrystallization.

### 2.3. Spectroscopic Characterization

(*E*)-2-Benzoyl-3-[4-(pyrrolidin-1-yl)phenyl]acrylonitrile (**4a**): 223 mg (74% yield). Orange crystals, mp 156–157 °C. IR (ATR, cm$^{-1}$): 2955, 2924, 2866, 2191, 1643, 1605, 1551, 1504, 1447, 1404, 1366, 1346, 1277, 1231, 1180, 1165, 1115, 1057, 1034, 964, 934, 860, 822, 795, 710, 694. $^1$H NMR (CDCl$_3$): 2.04–2.08 (m, 4H, 2CH$_2$), 3.40–3.44 (m, 4H, 2CH$_2$N), 6.59 (d, 2H, *J* = 8.9 Hz, Ar), 7.45–7.49 (m, 2H, Ar), 7.53–7.57 (m, 1H, Ar), 7.82 (d, 2H, *J* = 8.9 Hz, Ar), 7.96–8.00 (m, 3H, Ar, CH=CCN). $^{13}$C NMR (CDCl$_3$): 25.4 (2CH$_2$), 48.0 (2CH$_2$), 101.0 (<u>C</u>CN), 112.3 (2CH), 119.5 (C), 119.7 (C), 128.4 (2CH), 129.0 (2CH), 132.2 (CH), 134.9 (2CH), 137.6 (C), 151.8 (C–N), 155.9 (<u>C</u>H=CCN), 190.3 (C=O). HRMS (ESI) *m/z*: [M + H]$^+$ calcd. for C$_{20}$H$_{19}$N$_2$O: 303.1497, found 303.1496.

All of the spectral data of other compounds can be found in the Supplementary Materials.

## 3. Results and Discussion

In the context of our general interest in the development of new multicomponent reactions [27–33], we investigated three-component condensation of β-ketonitriles 1, 4-fluorobenzaldehyde 2, and secondary cyclic amines 3 (in a ratio of 1:1:2) (Scheme 1). All reactions were conducted in acetonitrile at reflux temperature for 6 h. As methylene active nitriles, acetonitriles containing benzoyl, pivaloyl, and 1-adamantanoyl groups were used. In each case, smooth reactions occurred to generate desired products 4a–i in good yields (63–75%). The practicality of this approach was demonstrated in the relatively large-scale synthesis of 4a from 10 mmol of benzoylacetonitrile, which was obtained in 76% yield compared to a 74% yield for 1 mmol of nitrile. Furthermore, nitriles with heteroaromatic substituents at the carbonyl group such as pyrrol-2-yl and indol-3-yl were also investigated, affording α-arylidene-β-ketonitriles **4j**–**m** in 79–90% yields. As secondary cyclic amines, we utilized pyrrolidine, morpholine, piperazine, ethyl piperazine-1-carboxylate, and 6-methoxy-2,3,4,9-tetrahydro-1*H*-pyrido[3,4-*b*]indole. In order to establish the generality of this three-component reaction, we extended the above method to β-carbonyl substituted nitriles with 2,3,4,9-tetrahydro-1*H*-carbazole and 2-nitroaniline fragments. To our satisfaction, these reactions proceeded efficiently to access the α-arylidenenitriles 4n,o bearing amide functional group. Products can be easily purified from impurities by single recrystallization, and chromatographic purification is not usually required. However, we failed to introduce 2-fluoro-, 4-chloro-, and 4-bromobenzaldehydes into the reaction. In boiling acetonitrile, for example, the reaction of 1-adamantanoylacetonitrile or benzoylacetonitrile with 2-fluorobenzaldehyde and pyrrolidine did not proceed. Apparently, this is due to steric difficulties in the formation of intermediate Meisenheimer complex.

**Scheme 1.** Scope of the reaction of β-carbonyl substituted nitriles **1** with 4-fluorobenzaldehyde **2** and secondary cyclic amines **3**.

The structures of the prepared compounds were confirmed by their IR, $^1$H, and $^{13}$C NMR spectral data and high-resolution mass spectra. In the IR spectra of β-ketonitriles **4a–m**, the absorption band of the cyano group appears at 2191–2207 cm$^{-1}$. In the $^{13}$C NMR spectra of compounds **4a–m**, the carbon atom of the C=O group resonated in the 175.6–199.1 ppm range, and the carbon atom of the CN group was observed at 118.6–121.7 ppm. The strong polarization of the exocyclic C=C bond, the push–pull nature of which is due to the presence of a strong electron-donating (4-R$_2$NC$_6$H$_4$) and electron-withdrawing (CO, CN) groups at both carbon atoms of the C=C bond, is noteworthy. The signal of the carbon atom bonded to the electron-withdrawing group was detected at 98.0–104.0 ppm while the signal of the neighboring carbon atom bound to the aryl group appeared in the region of 153.5–156.4 ppm. In the $^1$H NMR spectra of compounds **4a–m**, protons of the CH group bonded to the aryl fragment appear as singlets at 7.94–8.25 ppm. The number of protons that were directly linked to $^{13}$C atoms, inferred from DEPT spectra, was in accordance with the presented structures. The most characteristic signals in the $^1$H and $^{13}$C NMR spectra of compounds **4a–m** are shown in Figure 2.

**Figure 2.** The characteristic signals in the $^1$H (red) and $^{13}$C (blue) NMR spectra of compounds **4a–m** (δ, ppm).

The (*E*) geometry of the double bonds was determined by comparison with known compounds [34–36]. Furthermore, we were able to confirm the configuration of the double bond of the α-arylidene-β-ketonitriles by measuring the carbon–proton coupling constants. For example, in the proton coupled $^{13}$C NMR spectrum of the compound **4f**, the nitrile carbon atom appeared as a doublet at 120.6 ppm with the carbon–proton coupling constant $^3J_{CH}$ = 12.8 Hz corresponding to H,CN coupling. $^3J_{CH}$ could also be determined by conducting the HMBC experiment. In the HMBC spectrum of **4f**, the $^1$H δ/$^{13}$C δ crosspeak at 8.13/120.6 ppm corresponded to H,CN coupling ($^3J_{CH}$ = 12.8 Hz) (Figure 3). This is consistent with the *trans* $^3J_{CH}$ couplings reported in the literature [36,37]. The $^3J_{CH}$ for a *cis* relationship was approximately 8.5 Hz.

For the proposed three-component reaction, two main reaction pathways are possible. In the first route, the Knoevenagel condensation proceeds first and is followed by the aromatic nucleophilic substitution of the fluorine atom, which proceeds through the Meisenheimer complex. Alternatively, the S$_N$Ar process occurs prior to the Knoevenagel condensation [38]. An additional experiment showed that the substitution of the fluorine

atom in 4-fluorobenzaldehyde in boiling acetonitrile proceeded much more slowly than the condensation with β-ketonitrile. Therefore, when heating 4-fluorobenzaldehyde with 2 equiv. of pyrrolidine for 15 h, 4-pyrrolidinobenzaldehyde described in the literature was isolated only in 30% yield. Therefore, the first reaction pathway is more likely. The presumable reason for the Knoevenagel condensation and then subsequent S$_N$Ar process is that the Knoevenagel condensation product is more reactive than 4-fluorobenzaldehyde for the S$_N$Ar substitution due to its strong electron-withdrawing ability in addition to the resonance effect. At the same time, neither the bromine nor chlorine atoms are sufficiently activated to the S$_N$Ar process.

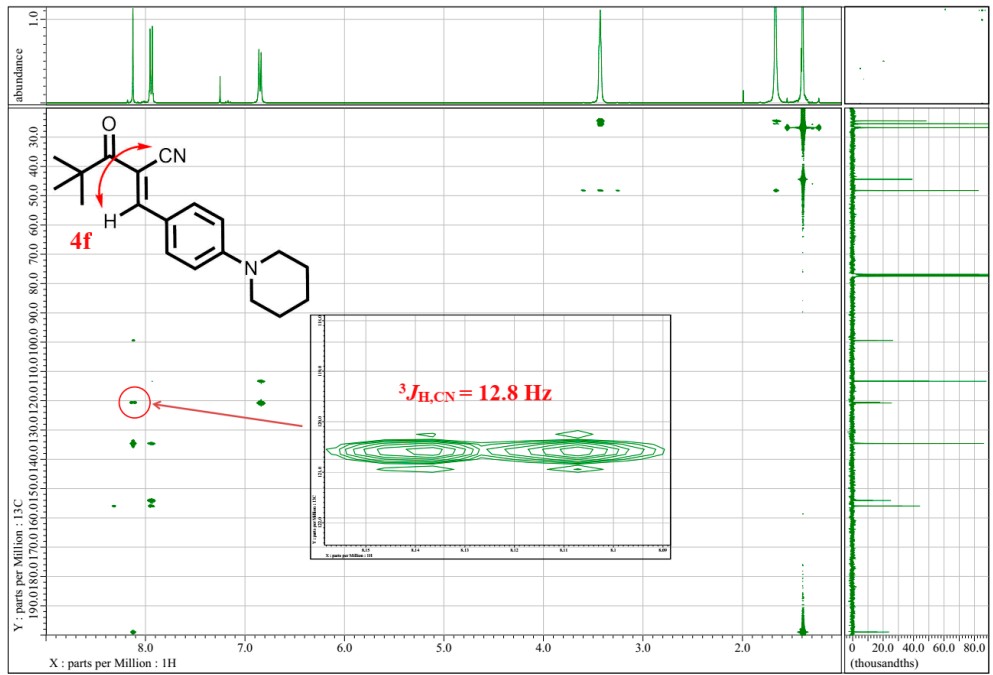

**Figure 3.** HMBC determination of the stereochemistry of α-arylidene-β-ketonitrile **4f**.

In order to broaden the scope of this process, we extended the study to the substate **5** bearing the 6-amino-1,3-dimethyluracil moiety. However, it turned out that in the reaction of β-ketonitrile **5** with 4-fluorobenzaldehyde and pyrrolidine or morpholine, instead of the expected products **6** of the cascade transformation including Knoevenagel condensation and aromatic nucleophilic substitution, α-arylidene-β-ketonitriles **7a,b** containing a fluorine atom were formed (Scheme 2). Apparently, in this case, the Knoevenagel condensation also occurred first, followed by the conjugated addition of a secondary amine, and the oxidation of the Michael adduct with atmospheric oxygen. In the $^{13}$C NMR spectra of the enaminonitriles **7a,b**, the carbon atoms of the benzene ring appeared as doublets due to splitting on the fluorine atom. In the $^{19}$F NMR spectra, signals of fluorine atoms were detected at −111.5 ppm.

To further expand the scope of the substrates, we tried to introduce 3-(dicyanomethylidene) indan-1-one **8**, which is easily available from indane-1,3-dione and malononitrile [39], into the three-component condensation with 4-fluorobenzaldehyde and pyrrolidine. However, in this case, the nucleophilic substitution of the fluorine atom also did not occur. According to the spectral data, the isolated product was the 9*H*-indeno[2,1-*c*]pyridin-9-one derivative **9**. In the $^{19}$F NMR spectrum, the fluorine atom appeared at −109.8 ppm. In the $^{13}$C NMR spectrum, the characteristic signal at 114.6 ppm referred to the carbon atom of the nitrile group. The carbonyl carbon atom resonated at 188.0 ppm and the carbon atom bound to fluorine appeared as a doublet signal at 164.4 ppm ($^1J_{CF}$ = 248.9 Hz). We assumed that the product formation started with a normal Knoevenagel reaction between indanone **8** and 4-fluorobenzaldehyde **2**, followed by a nucleophilic attack of pyrrolidine at the CN

group. It is an open question whether the thus generated zwitterionic imidide **A** is trapped in a concerted manner by the present enone part to form the zwitterionic enolate **B**, which then yields 1,9a-dihydro-9*H*-indeno[2,1-*c*]pyridin-9-one **C** by the intermolecular H$^+$ shift, or whether the imidide **A** has a certain life-time, so that it can undergo intermolecular H$^+$ shift to the neutral 1-azahexatriene intermediate **D**, which then experiences an electrocyclic disrotatory ring closure to 1,9a-dihydro-9*H*-indeno[2,1-*c*]pyridin-9-one **C** [40]. In any case, subsequent aerobic oxidation of the intermediate **C** resulted in a [5+1]-cyclocondensation product **9** in 68% yield (Scheme 3).

**Scheme 2.** 3-(6-Amino-1,3-dimethyl-2,4-dioxo-1,2,3,4-tetrahydropyrimidin-5-yl)-3-oxopropanenitrile **5** in the three-component reaction.

**Scheme 3.** Synthesis of the 9*H*-indeno[2,1-*c*]pyridin-9-one derivative **9**.

Having established a strategy for the synthesis of 4-aminobenzylidene derivatives of β-ketonitriles, the applicability of these structures was studied. We have shown that the interaction of α-arylidene-β-ketonitriles **4g,i** with malononitrile and ethyl cyanoacetate in the presence of catalytic amounts of piperidine in refluxing ethanol results in a rapid formal substitution of the β-ketonitrile fragment for the residues of the methylene active nitriles used. The mechanism of the reaction involves consecutive carbo- and retro-carbo-Michael reactions (Scheme 4).

R₂N = morpholino, EWG = CN (**10a**, 82%, from **4g**)
R₂N = pyrrolidino, EWG = CO₂Et (**10b**, 80%, from **4i**)

**Scheme 4.** Reaction of α-arylidene-β-ketonitriles **4g,i** with methylene active nitriles.

The reaction of equimolar amounts of pyridinium salt **11** and ketonitrile **4b** in the presence of triethylamine (1 equiv) led to the formation of 5-(4-methoxybenzoyl)-2-phenyl-4-[4-(piperidin-1-yl)phenyl]-4,5-dihydrofuran-3-carbonitrile **12** as individual *trans*-isomers in 76% yield (Scheme 5). The reaction was carried out in MeCN under reflux under an argon atmosphere for 5 h.

**Scheme 5.** Synthesis of *trans*-5-(4-methoxybenzoyl)-2-phenyl-4-[4-(piperidin-1-yl)phenyl]-4,5-dihydrofuran-3-carbonitrile **12**.

In the $^1$H NMR spectra of **12**, protons H-4 and H-5 in the dihydrofuran ring appeared as doublets at 4.59 and 5.84 ppm, respectively, with a coupling constant of $^3J$ = 5.7 Hz, which corresponds to the *trans*-isomer according to the literature [28,41]. A mechanism of the formation of dihydrofuran **12** includes the conjugated addition of pyridinium ylide generated in situ from the pyridinium salt **11** under the action of base with ketonitrile **4b** to give the zwitterionic intermediate **E**, which undergoes 5-*exo-tet*-cyclization to the dihydrofuran **12**. The final step is a classic intramolecular $S_N2$ substitution reaction that necessitates a nucleophilic attack by enolate **E** from the back side of the electrophilic carbon atom bearing the leaving pyridinium group. Due to steric hindrance in the transition state, the 5-acyl and 4-aryl groups occupy the *trans*-position relative to each other.

## 4. Conclusions

In summary, we developed a new three-component Knoevenagel–nucleophilic aromatic substitution reaction of β-ketonitriles, 4-fluorobenzaldehyde, and cyclic secondary amines. This process affords the one-pot formation of one carbon–nitrogen bond and one carbon–carbon double bond from simple and readily accessible reagents in high ef-

ficiency. All of the reactants were taken in their stoichiometric ratio. The good yields of 4-aminobenzylidene derivatives of β-ketonitriles clearly exemplify the potential of the reported methodology. The reaction can also be extended to α-cyanoamides. Taking into account the high importance of α-arylidene-β-ketonitriles, our method might be beneficial to prepare these compounds in a convenient way in synthetic and medicinal chemistry. Furthermore, the developed method can be used in the synthesis of more complex compounds including those used as intermediates in the pharmaceutical industry.

**Supplementary Materials:** The following supporting information can be downloaded at: https://www.mdpi.com/article/10.3390/reactions3040042/s1, Physical and NMR data of all products.

**Author Contributions:** Conceptualization, D.V.O. and V.A.O.; Methodology, D.V.O.; Investigation and data collection, D.V.O. and K.S.K.; Writing—original draft preparation, K.S.K. and V.A.O.; Writing—review and editing, K.S.K. and V.A.O.; Supervision, D.V.O.; Project administration, D.V.O.; Funding acquisition, V.A.O. All authors have read and agreed to the published version of the manuscript.

**Funding:** This research was funded by the Russian Science Foundation (grant 22-13-00253).

**Data Availability Statement:** Not applicable.

**Acknowledgments:** We thank the Center for Collective Use "Investigation of the physicochemical properties of substances and materials" of Samara State Technical University for their assistance with the NMR spectroscopy.

**Conflicts of Interest:** The authors declare no conflict of interest.

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
