# Peer review of "Three-Component Condensation of β-Ketonitriles, 4-Fluorobenzaldehyde, and Secondary Cyclic Amines"

_reactions, doi:10.3390/reactions3040042_

Round 1

Reviewer 1 Report

Dmitry V. Osipov et al., report a new one-pot three-component condensation of β-ketonitriles, 4-fluorobenzaldehyde and secondary cyclic amines. The authors have demonstrated that this process affords one-pot formation of one carbon–nitrogen bond and one carbon–carbon double bond from simple and readily accessible reagents in high efficiency.

The work presented in this manuscript is scientifically interesting; it describes a new approach for the synthesis of α-arylidene-β-Ketonitriles by using one-pot three-component condensation. The protocol developed here appears clearly as an improvement of existing preparation of these products. The description of these syntheses and the characterization of these compounds are well explained.

Overall the manuscript is rich and interesting; and the paper structure is well-knit and suitable for publication in the journal, after minor revisions. The comments are listed as the following points:

1-    Some corrections should be made (lack of space, forgotten points to add or others to delete)? to check.

2-    In introduction, line 33, “as anti-hyperglycemic I [3] and anti-tuberculosis II [4] agents, entacapone III [5]” should be “as anti-hyperglycemic (compound I) [3] and anti-tuberculosis (compound II) [4] agents, entacapone (Compound III) [5]”.

3-    In introduction, line 36, “(for instance, IV-VI) were identified (Figure 1)” should be “(for instance, Compounds IV-VI) were identified (Figure 1)”.

4-    Line 70, “4a-c, e-o” and “7a,b, 9” should be in bold.

5- Besides, authors should provide HRMS images (in Supplementary data) for purity verification.

Author Response

2. fixed
3. fixed
4. Bold font in the subtitle is automatically corrected to non-bold
5. Unfortunately, the original files were lost due to a recent breakdown of the HRMS computer. We also didn't see the requirement to bring copies in the rules. We can carry out elemental analysis and give its data within a week or remake HRMS within a month due to instrument setting

Reviewer 2 Report

The reviewed work concerns the three-component condensation of β-ketonitriles, 4-fluorobenzaldehyde, and secondary cyclic amines. The research is well-planned and the results are documented/confirmed. I can recommend this work for publication in Reactions after some minor corrections:

- are any of the compounds 4 known from the literature?

- page 5, lines 142-155 - the Authors describe two possible pathways - an appropriate scheme should be provided

- description ~H+ is strange I would use -H+ instead

- Scheme 4: I suppose, one step is missed, otherwise the charges do not match

- how did you determine the configuration (E/Z) of compounds 10a-b (based on literature?)

- descriptions of compounds 10a and 10b were mixed; It should be: R2N - morpholine, EWG = CO2Et: 10a and R2N - pyrrolidino, EWG = CN: 10b - please check it.

Supporting:

- compound 4m, 6.57 ppm - please correct "Hz"

- compounds 7a and 7b - 19F NMR spectra require phase correction.

Author Response

- are any of the compounds 4 known from the literature?
mentioned compounds have not been previously described
- page 5, lines 142-155 - the Authors describe two possible pathways - an appropriate scheme should be provided
we consider the presence of the mechanisms of the Knoevenagel reaction and SNAr as a truism, since they are given in every student textbook
- description ~H+ is strange I would use -H+ instead
we used the standard designation (~H+) for proton transfer process
- Scheme 4: I suppose, one step is missed, otherwise the charges do not match
scheme has been corrected
- how did you determine the configuration (E/Z) of compounds 10a-b (based on literature?)
the configuration is determined by analogy with compounds 4
- descriptions of compounds 10a and 10b were mixed; It should be: R2N - morpholine, EWG = CO2Et: 10a and R2N - pyrrolidino, EWG = CN: 10b - please check it.
we have given the correct names for mentioned substituents

Supporting :
- compound 4m, 6.57 ppm - please correct "Hz"
(has been corrected)
- compounds 7a and 7b - 19F NMR spectra require phase correction 

(has been corrected)

Reviewer 3 Report

Manuscript by Osyanin et al. reports the three-component condensation β-ketonitriles, 4-fluorobenzaldehyde and secondary cyclic amines. The substrate scope of reaction is studied well. They also found that in case of 6-amino-1,3-dimethyluracil moiety fluorine substituted product was not formed and they got unexpected michael addition product. This work is well done. The protocol is synthetically attractive. Most of the spectra are good quality. I would recommend the manuscript for publication in Reactions after minor revision.

1. The authors are encouraged to probe the scalability of the process.

2. They should provide the reason why reaction did not proceed with 2-fluorobenzaldehyde.

3. In case of 2-fluorobenzaldehyde, did they get Knoevenagel adduct or they recovered the starting material?

Author Response

1. The authors are encouraged to probe the scalability of the process 

The sentence "The practicality of this approach was demonstrated in the relatively large-scale synthesis of 4a from 10 mmol of benzoylacetonitrile, which was obtained in 76% yield compared to a 74% yield for 1 mmol of nitrile" has been added

2. They should provide the reason why reaction did not proceed with 2-fluorobenzaldehyde.

The sentence "Apparently this is due to steric difficulties in the formation of intermediate Meisenheimer complex." has been added 

3. In case of 2-fluorobenzaldehyde, did they get Knoevenagel adduct or they recovered the starting material?

the reaction interrupted at the knoevenagel product formation stage